# Sex differences in the association between systemic oxidative stress status and optic nerve head blood flow in normal-tension glaucoma

**Masataka Sato**[1], **Masayuki Yasuda**[1], **Nana Takahashi**[1], **Kazuki Hashimoto**[1], **Noriko Himori**[1,2], **Toru Nakazawa**[1,3,4,5] *

1 Department of Ophthalmology, Tohoku University Graduate School of Medicine, Miyagi, Japan, 2 Department of Aging Vision Healthcare, Tohoku University Graduate School of Biomedical Engineering, Miyagi, Japan, 3 Department of Ophthalmic Imaging and Information Analytics, Tohoku University Graduate School of Medicine, Miyagi, Japan, 4 Department of Retinal Disease Control, Tohoku University Graduate School of Medicine, Miyagi, Japan, 5 Department of Advanced Ophthalmic Medicine, Tohoku University Graduate School of Medicine, Miyagi, Japan

☯ These authors contributed equally to this work.
* ntoru@oph.med.tohoku.ac.jp

## Abstract

### Purpose

To investigate the association of systemic oxidative stress markers and optic nerve head (ONH) blood flow in normal-tension glaucoma (NTG) patients, as well as sex differences in this association.

### Methods

This was a cross-sectional study of 235 eyes with NTG of 134 patients (56 male, 78 female; mean age, 60.9±14.1 years). Laser speckle flowgraphy (LSFG) was used to measure ONH blood flow (mean blur rate in the tissue area of the ONH; MBR-T) and LSFG pulse-waveform parameters, including flow acceleration index in the tissue area of the ONH (FAI-T). Oxidative stress markers, diacron-reactive oxygen metabolites (d-ROMs), and biological antioxidant potential (BAP) were measured with a free radical elective evaluator. Spearman's rank correlation test and a multivariate linear mixed-effect model were used to investigate factors associated with ONH blood flow.

### Results

MBR-T was significantly correlated with age (rs = -0.28, p < 0.001), mean arterial pressure (rs = -0.20, p = 0.002), intraocular pressure (rs = 0.24, p < 0.001), peripapillary retinal nerve fiber layer thickness (rs = 0.62, p < 0.001), and disc area (rs = -0.26, p < 0.001), but not with serum d-ROM level. Separate analyses of the subjects divided by sex showed that BAP was positively correlated to MBR-T (rs = 0.21, p = 0.036) and FAI-T (rs = 0.36, p < 0.001) only in male subjects. Similarly, BAP was significantly associated with MBR-T (β = 0.25,

**Data Availability Statement:** All relevant data are within the manuscript and its Supporting information files.

**Funding:** This paper was supported in part by JSPS KAKENHI Grants-in-Aid for Scientific Research (B) (T.N. JP20H03838), Grant-in-Aid for Challenging Exploratory Research (T.N. JP21K19548), and Grant-in-Aid for Early-Career Scientists (M.Y. JP21K16866).The funders had no role in study design, data collection and analysis, decision to publish, or preparation of the manuscript.

**Competing interests:** The authors have declared that no competing interests exist.

$p = 0.026$) and FAI-T ($\beta = 0.37$, $p < 0.001$) in male subjects in a multivariate linear mixed-effect model.

## Conclusion

A lower serum antioxidant level, as indicated by BAP, was associated with reduced ONH blood flow only in male NTG patients. Our findings suggest that there are sex differences in the involvement of oxidative stress in the pathogenesis of reduced ocular blood flow in NTG.

## Introduction

Glaucoma is the leading cause of blindness worldwide, with a prevalence that is expected to increase to 111.8 million by 2040 [1]. It is an ocular neurodegenerative disease characterized by progressive retinal ganglion cell death [2]. Elevated intraocular pressure (IOP) is the major risk factor for glaucoma, and the most common evidence-based treatment for glaucoma is lowering IOP [3]. However, we have often observed patients who have progressive glaucoma despite this treatment in daily clinical practice [4]. Normal-tension glaucoma (NTG) is defined as glaucoma with an open angle and no record of IOP 22 mmHg or higher. Epidemiological research has shown that the proportion of NTG cases among all cases of primary open-angle glaucoma was 50% in the US and 75 to 90% in China, Singapore, Japan, and Korea [5]. In Japan, NTG is the most common subtype of open-angle glaucoma, where it accounts for over 90% of cases [6]. Like other neurodegenerative diseases, it is considered that multiple factors are implicated in the pathogenesis of NTG [7]. However, IOP-independent risk factors for NTG remain unclear. In order to establish a novel treatment for glaucoma, there is a need to identify these risk factors.

Blood flow impairment is involved in various age-related diseases, such as Alzheimer's disease [8] and major depressive disorder [9]. Many studies have suggested that changes in ocular blood flow (OBF) play a critical role in the development and progression of glaucoma. Blood flow in the peripapillary retina, which can be measured with laser Doppler flowmetry, has been found to be reduced in NTG [10]. Blood flow in the capillary area of the ONH, which can be measured with laser speckle flowgraphy (LSFG), has been shown to be reduced in the pre-perimetric stage of glaucoma [11] and the early stage of NTG [12]. In addition, the severity of glaucoma has been reported to be correlated to decreased ONH blood flow [13]. Furthermore, we recently demonstrated that decreased tissue blood flow in the ONH precedes glaucoma neurodegeneration in patients with risk factors for altered blood flow, such as old age and high pulse rate [14]. Treatment to inhibit reduced ONH blood flow may prevent the progression of glaucomatous optic neuropathy, but such treatment has not been established, nor have biomarkers for ONH blood flow in glaucoma.

Oxidative stress is defined as a state of an excess load of reactive oxygen species and/or reduction of antioxidants in cells and tissues. Chronic oxidative stress is involved in various age-related diseases, such as cancer [15] and atherosclerotic diseases [16]. Previous studies have suggested that systemic oxidative stress plays an important role in the pathogenesis of glaucoma. Himori et al. reported that urine levels of 8-hydroxy-2'-deoxyguanosine (8-OHdG), a systemic oxidative stress marker, were correlated to visual field defects in NTG [17]. In addition, Tanito et al. measured both systemic oxidative stress and the antioxidant capacity of blood samples using a free-radical analyzer system and found a positive correlation between antioxidant potential and mean deviation of the visual field [18]. Oxidative stress can induce

decreased nitric oxide release and increased endothelin-1 production, which is involved in the blood flow impairment underlying various vascular diseases [19]. We thus hypothesized that systemic oxidative stress levels were associated with a reduction of ocular blood flow in NTG. In the current study, we aimed to investigate the relationship between markers of systemic oxidative stress and ONH blood flow, measured with LSFG, in NTG patients.

## Materials and methods

### Subjects

This was a cross-sectional study that examined 235 eyes of 134 NTG patients who visited Tohoku University Hospital, located in Miyagi, Japan, between October 2018 and July 2020. Written informed consent was obtained from the subjects at the time of sample collection. This study followed the tenets of the Declaration of Helsinki and was approved by the Ethics Committee of Tohoku University School of Medicine (Protocol number: 2021-1-430). NTG was diagnosed by a glaucoma specialist (T.N.) and was defined as an open angle in a gonioscopic examination, glaucomatous ONH changes, and corresponding visual field defects matching the Anderson–Patella criteria [20]. The exclusion criteria were as follows: (1) presence of ocular disease other than mild cataract, (2) a history of high IOP (22 mmHg or higher, as measured with Goldmann applanation tonometry) on the test day, and (3) high myopia, i.e., axial length (AL) longer than 26 mm.

### Ophthalmic evaluation

Intraocular pressure (IOP) was measured with Goldmann applanation tonometry; AL was measured with the IOL Master (Zeiss Meditec, Dublin, CA, USA); circumpapillary retinal nerve fiber layer thickness (cpRNFLT) was measured with swept-source OCT (DRI OCT Triton, Topcon, Inc., Tokyo, Japan), and the visual field was measured with the 24–2 program of the Humphrey Field Analyzer (Carl Zeiss Meditec, Dublin, CA, USA). When both eyes of a patient met the inclusion criteria, both eyes were included in the statistical analysis.

### Blood samples and oxidative stress measurement

The presence or absence of smoking history, diabetes (DM), hypertension, and dyslipidemia were recorded before the samples were collected. Blood samples were collected after at least 3 hours of fasting. Next, oxidative stress markers, including diacron-reactive oxygen metabolites (d-ROMs) and biological active potential (BAP), were measured with a free radical analyzer (Free Carpe Diem, Wismerll Co., Ltd., Tokyo, Japan). D-ROM level reflects oxidative stress and is an indicator of the serum activity of hydroperoxides. BAP reflects the antioxidant potential of a sample and represents its capacity to reduce ferric oxide to ferrous oxide. Details of these analysis procedures have been described previously [21].

### Optic nerve head blood flow assessment with laser speckle flowgraphy

ONH blood flow was measured with LSFG (LSFG-NAVI, Softcare Co., Ltd., Fukutsu, Japan), as previously described [22]. Before the measurement, the pupils of the eyes of each subject were dilated with 0.5% tropicamide. Systolic and diastolic blood pressure (SBP and DBP) and pulse rate (PR) were measured after the subjects had rested for 10 min in a sitting position in a dark room. Mean arterial pressure (MAP) was calculated as follows: MAP = diastolic BP (DBP) + 1/3 (systolic BP [SBP]—DBP) [22]. Following these measurements, color maps of mean blur rate (MBR) in the ONH were obtained with LSFG, which is derived from the speckle pattern produced by the interference of a laser scattered by moving blood cells [23].

The MBR color maps were then automatically divided by the LSFG software (LSFG analyzer, version 3.2.12.0) into separate regions of the ONH, comprising the large vessel and tissue areas. MBR was assessed separately in the vessel and tissue areas and in the ONH overall. Our study focused on MBR in the tissue area (MBR-T) of the ONH, because it is considered to be a good indicator of blood flow in the deep ONH [24]. Pulse-waveform parameters were also measured with LSFG in the tissue area of the ONH, including flow acceleration index (FAI-T), blowout score (BOS-T), and blowout time (BOT-T). FAI-T indicates the instantaneous force with which blood flow increases in a short time; it is calculated as the maximum change among all frames (1/30 s) in a rising curve [25]. BOS-T is an indicator of the volume of blood flow maintained in a vessel between heartbeats [26]. BOT-T is defined as the ratio of the half width (i.e., the time that the waveform is higher than half of the mean of the minimum and maximum signals) in one heartbeat [27,28]. All statistical analyses of ONH blood flow were based on an average of three separate LSFG measurements.

## Statistical analysis

All data are shown as the median (interquartile range [IQR]). Fisher's exact test was used to analyze the significance of differences in the variables between male and female groups. Spearman's correlation coefficient and a multivariate linear mixed-effect model were used to analyze the relationship of MBR-T and FAI-T to other variables. All statistical analyses were performed with R software version 4.1.1 (R Core Team 2021). The significance level was set at $p < 0.05$. All data were fully anonymized before analysis.

## Results

The clinical and ophthalmological characteristics of the NTG patients enrolled in this study are shown in Tables 1 and 2, respectively. The male patients were older ($p < 0.001$), had higher SBP ($p < 0.001$), higher DBP ($p = 0.016$), higher MAP ($p = 0.001$), a lower d-ROM level ($p = 0.014$), lower IOP ($p < 0.001$), a wider disc area ($p < 0.001$), lower MBR-T ($p < 0.001$), and lower FAI-T ($p = 0.004$). A history of hypertension was more frequent in the male patients

**Table 1. Systemic characteristics of the normal-tension glaucoma patients.**

| Variable | Overall (N = 134) | Male (N = 56) | Female (N = 78) | p |
|---|---|---|---|---|
| Age (years) | 62.0 (52.0–70.0) | 67.5 (60.0–74.0) | 57.5 (47.5–67.0) | <0.001*** |
| SBP (mmHg) | 124.0 (114.3–136.0) | 129.0 (121.0–140.0) | 119.0 (110.0–132.8) | <0.001*** |
| DBP (mmHg) | 73.0 (66.0–81.8) | 74.5 (68.8–85.0) | 71.5 (64.0–78.0) | 0.016* |
| MAP (mmHg) | 90.0 (83.4–100.6) | 93.2 (88.6–103.0) | 86.2 (80.0–96.0) | 0.001** |
| HR (bpm) | 69.0 (62.0–76.0) | 68.0 (60.0–75.3) | 70.0 (64.0–76.0) | 0.209 |
| d-ROM (U. Carr) | 379.5 (344.0–416.8) | 357.5 (333.8–404.5) | 391.0 (351.0–427.0) | 0.014* |
| BAP (μmol/L) | 2208.0 (2035.0–2313.5) | 2159.0 (2001.8–2301.3) | 2219.0 (2085.0–2313.5) | 0.419 |
| HT (n) | 39 | 25 | 14 | 0.001** |
| DM (n) | 12 | 5 | 7 | 0.993 |
| DL (n) | 34 | 14 | 20 | 1 |
| Smoking history (n) | 12 | 8 | 4 | 0.127 |

SBP: Systolic blood pressure; DBP: Diastolic blood pressure; MAP: Mean arterial pressure; HR: Heart rate; d-ROMs: Diacron reactive oxygen metabolites; U. Carr: Carrelli units; BAP: Biological antioxidant potential; HT: Hypertension; DM: Diabetes; DL: Dyslipidemia. Data are expressed as the median (IQR). The Mann-Whitney U test was used to compare groups. Categorical variables were analyzed with Fisher's exact test. Asterisks indicate statistical significance (*: $p < 0.05$, **: $p < 0.01$, ***: $p < 0.001$).

**Table 2. Ocular characteristics of normal-tension glaucoma patients.**

| Variable | Overall (N = 235) | Male (N = 100) | Female (N = 135) | p |
|---|---|---|---|---|
| IOP (mmHg) | 12.00 (11.00–14.00) | 12.00 (10.00–14.00) | 13.00 (12.00–15.00) | <0.001*** |
| Axial length (mm) | 24.65 (23.87–25.40) | 24.42 (23.98–25.03) | 24.71 (23.80–25.51) | 0.355 |
| cpRNFLT (μm) | 65.78 (53.55–78.66) | 62.45 (47.00–77.08) | 67.03 (56.50–81.12) | 0.046* |
| Disc area (mm$^2$) | 1.96 (1.68–2.29) | 2.12 (1.77–2.44) | 1.87 (1.58–2.12) | <0.001*** |
| MBR-T (AU) | 9.07 (7.53–11.13) | 8.32 (6.80–10.01) | 9.73 (8.23–11.73) | <0.001*** |
| FAI-T (AU) | 0.97 (0.73–1.37) | 0.88 (0.63–1.27) | 1.07 (0.80–1.38) | 0.003** |
| BOS-T | 77.67 (73.87–81.62) | 77.73 (73.60–81.77) | 77.67 (73.92–81.58) | 0.89 |
| BOT-T | 50.37 (47.30–53.58) | 49.63 (46.41–53.20) | 50.80 (47.52–53.68) | 0.133 |
| Topical anti-glaucoma drugs (n) | 3.00 (2.00–4.00) | 3.00 (2.00–4.00) | 3.00 (2.00–4.00) | 0.912 |

IOP: Intraocular pressure; cpRNFLT: Circumpapillary retinal nerve fiber layer thickness; MBR-T: Mean blur rate in the tissue area of the optic nerve head; FAI-T: Flow acceleration index in the tissue area of the optic nerve head; AU: Arbitrary units; BOS-T: Blowout score in the tissue area of the optic nerve head; BOT-T: Blowout time in the tissue area of the optic nerve head. Data are expressed as the median (IQR). The Mann-Whitney U test was used to compare groups. Asterisks indicate statistical significance (*: p < 0.05, **: p < 0.01, ***: p < 0.001).

(p = 0.001). The number of topical anti-glaucoma drugs was not significantly different between the sexes (p = 0.912).

Spearman's rank correlation coefficient was calculated to evaluate the relationships between ONH blood flow parameters and their respective clinical parameters (Table 3). MBR-T was significantly correlated with age (rs = -0.28, p < 0.001), MAP (rs = -0.20, p = 0.002), IOP (rs = 0.24, p < 0.001), cpRNFLT (rs = 0.62, p < 0.001), and disc area (rs = -0.26, p < 0.001) in all patients. Separate analyses of patients grouped by sex showed that MBR-T was correlated to age (rs = -0.27, p = 0.002), MAP (rs = -0.21, p = 0.016), IOP (rs = 0.27, p = 0.002), and disc area (rs = -0.27, p = 0.002) in the female subjects, but not in the male subjects (age, rs = -0.10, p = 0.306; MAP, rs = 0.05, p = 0.602; IOP, rs = 0.006, p = 0.956; disc area, rs = -0.13, p = 0.193). We also investigated the correlation between the OBF parameters (i.e., MBR-T and FAI-T) and systemic oxidative stress markers (BAP and d-ROMs) (Table 3 and S1 Fig). MBR-T was significantly correlated to BAP only in the male patients (rs = 0.21, p = 0.036). There was no significant correlation between MBR-T and d-ROM level in any group. FAI-T was significantly correlated with age (rs = -0.13, p = 0.050), MAP (rs = -0.25, p < 0.001), IOP (rs = 0.21, p = 0.001), cpRNFLT (rs = 0.54, p < 0.001), BAP (rs = 0.16, p = 0.017), and disc area (rs = -0.21, p = 0.010). Analyses by sex showed that age (rs = -0.18, p = 0.041), MAP (rs = -0.30, p < 0.001), d-ROM level (rs = -0.18, p = 0.036), IOP (rs = 0.22, p = 0.010), and disc area (rs = -0.17, p = 0.045) were correlated to FAI-T only in the female subjects. CpRNFLT was significantly correlated to FAI-T in both sexes. FAI-T was correlated to BAP only in the male subjects (rs = 0.36, p < 0.001). BOS-T was significantly correlated with age (rs = -0.31, p < 0.001), MAP (rs = 0.17, p = 0.010), HR (rs = 0.28, p < 0.001), and cpRNFLT (rs = -0.24, p < 0.001). BOT-T was significantly correlated with age (rs = -0.44, p < 0.001) and HR (rs = 0.35, p < 0.001).

Multivariate linear mixed-effect models were used to evaluate the association of BAP with MBR-T and FAI-T while adjusting for potential confounding factors, including sex, age, IOP, cpRNFLT, disc area, MAP, BAP, smoking history, HT, DM, and DL (Table 4). CpRNFLT was significantly associated with MBR-T in both sexes and in the group overall (overall, β = 0.45, p < 0.001; male, β = 0.51, p < 0.001; female, β = 0.44, p < 0.001). Disc area was also associated with MBR-T in both sexes and in the group overall (overall, β = -0.16, p = 0.003; male, β = -0.21, p = 0.021; female, β = -0.15, p = 0.028). This finding is compatible with a previous study conducted with healthy subjects [25]. Although IOP was associated with MBR-T in the overall

**Table 3. Spearman's correlation coefficient between MBR-T, pulse waveform parameters, and each variable.**

| LSFG parameter | Variable | Overall | | Male | | Female | |
|---|---|---|---|---|---|---|---|
| | | rs | p | rs | p | rs | p |
| MBR-T (AU) | | | | | | | |
| | Age (years) | -0.28 | <0.001*** | -0.1 | 0.306 | -0.27 | 0.002** |
| | MAP (mmHg) | -0.2 | 0.002** | 0.05 | 0.602 | -0.21 | 0.016* |
| | HR (bpm) | 0.11 | 0.1 | -0.05 | 0.605 | 0.15 | 0.08 |
| | BAP (µmol/L) | 0.13 | 0.05 | 0.21 | 0.036* | 0.01 | 0.882 |
| | d-ROMs (U. Carr) | 0.008 | 0.901 | -0.03 | 0.766 | -0.09 | 0.283 |
| | IOP (mmHg) | 0.24 | <0.001*** | 0.006 | 0.956 | 0.27 | 0.002** |
| | cpRNFLT (µm) | 0.62 | <0.001*** | 0.6 | <0.001*** | 0.65 | <0.001*** |
| | Disc area (mm$^2$) | -0.26 | <0.001*** | -0.13 | 0.193 | -0.27 | 0.002** |
| FAI-T (AU) | | | | | | | |
| | Age (years) | -0.13 | 0.050* | 0.06 | 0.548 | -0.18 | 0.041* |
| | MAP (mmHg) | -0.25 | <0.001*** | -0.11 | 0.266 | -0.3 | <0.001*** |
| | HR (bpm) | 0.08 | 0.228 | 0.05 | 0.602 | 0.05 | 0.566 |
| | BAP (µmol/L) | 0.16 | 0.017* | 0.36 | <0.001*** | -0.07 | 0.437 |
| | d-ROMs (U. Carr) | -0.05 | 0.438 | 0.04 | 0.673 | -0.18 | 0.036* |
| | IOP (mmHg) | 0.21 | 0.001** | 0.06 | 0.544 | 0.22 | 0.010** |
| | cpRNFLT (µm) | 0.54 | <0.001*** | 0.53 | <0.001*** | 0.54 | <0.001*** |
| | Disc area (mm$^2$) | -0.21 | 0.010* | -0.16 | 0.112 | -0.17 | 0.045* |
| BOS-T | | | | | | | |
| | Age (years) | -0.31 | <0.001*** | -0.43 | <0.001*** | -0.27 | 0.001** |
| | MAP (mmHg) | 0.17 | 0.010* | 0.09 | 0.388 | 0.24 | 0.005** |
| | HR (bpm) | 0.28 | <0.001*** | 0.2 | 0.044* | 0.35 | <0.001*** |
| | BAP (µmol/L) | -0.01 | 0.82 | -0.16 | 0.104 | 0.13 | 0.135 |
| | d-ROMs (U. Carr) | 0.03 | 0.646 | -0.06 | 0.53 | 0.09 | 0.274 |
| | IOP (mmHg) | -0.11 | 0.104 | -0.11 | 0.262 | -0.12 | 0.166 |
| | cpRNFLT (µm) | -0.24 | <0.001*** | -0.24 | 0.017* | -0.25 | 0.004** |
| | Disc area (mm$^2$) | 0.06 | 0.388 | 0.06 | 0.523 | 0.03 | 0.691 |
| BOT-T | | | | | | | |
| | Age (years) | -0.44 | <0.001*** | -0.34 | <0.001*** | -0.51 | <0.001*** |
| | MAP (mmHg) | -0.11 | 0.087 | -0.08 | 0.388 | -0.09 | 0.296 |
| | HR (bpm) | 0.35 | <0.001*** | 0.38 | <0.001*** | 0.3 | <0.001*** |
| | BAP (µmol/L) | 0.08 | 0.224 | 0.06 | 0.523 | 0.1 | 0.252 |
| | d-ROMs (U. Carr) | -0.02 | 0.728 | -0.06 | 0.885 | -0.08 | 0.36 |
| | IOP (mmHg) | -0.11 | 0.104 | -0.11 | 0.268 | -0.09 | 0.309 |
| | cpRNFLT (µm) | -0.06 | 0.333 | -0.18 | 0.081 | -0.25 | 0.882 |
| | Disc area (mm$^2$) | -0.006 | 0.923 | 0.006 | 0.951 | 0.01 | 0.895 |

MBR-T: Mean blur rate in the tissue area of the optic nerve head; FAI-T: Flow acceleration index in the tissue area of the optic nerve head; AU: Arbitrary units; BOS-T: Blowout score in the tissue area of the optic nerve head; BOT-T: Blowout time in the tissue area of the optic nerve head; MAP: Mean arterial pressure; HR: Heart rate; BAP: Biological antioxidant potential; d-ROMs: Diacron reactive oxygen metabolites; U. Carr: Carrelli units; IOP: Intraocular pressure; cpRNFLT: Circumpapillary retinal nerve fiber layer thickness; rs: Spearman's rank correlation coefficient. Asterisks indicate statistical significance (*: p< 0.05, **: p < 0.01, ***: p < 0.001).

group of subjects and in the female subjects (overall, β = 0.11, p = 0.036; male, β = 0.07, p = 0.433; female, β = 0.15, p = 0.042), the β value was very small and the p value was close to the significance threshold. On the other hand, BAP was significantly associated with MBR-T only in the male subjects (overall, β = 0.03, p = 0.579; male, β = 0.24, p = 0.026; female,

**Table 4. The effect of clinical parameters on MBR-T and FAI analyzed with a linear mixed-effect model.**

| Response variable | Explanatory variable | Overall | | Male | | Female | |
|---|---|---|---|---|---|---|---|
| | | β | p | β | p | β | p |
| MBR-T (AU) | | | | | | | |
| | Sex | -0.1 | 0.155 | NA | NA | NA | NA |
| | Age (years) | -0.06 | 0.386 | 0.007 | 0.926 | -0.08 | 0.446 |
| | IOP (mmHg) | 0.11 | 0.036* | 0.07 | 0.433 | 0.15 | 0.042* |
| | cpRNFLT (μm) | 0.45 | <0.001*** | 0.51 | <0.001*** | 0.44 | <0.001*** |
| | Disc area | -0.16 | 0.003** | -0.21 | 0.021* | -0.15 | 0.028* |
| | MAP (mmHg) | -0.08 | 0.219 | 0.06 | 0.612 | -0.14 | 0.142 |
| | BAP (μmol/L) | 0.03 | 0.579 | 0.24 | 0.026* | -0.08 | 0.368 |
| | Smoking history | 0.07 | 0.252 | 0.04 | 0.693 | 0.06 | 0.505 |
| | HT | -0.09 | 0.183 | -0.21 | 0.046* | -0.01 | 0.905 |
| | DM | -0.06 | 0.364 | -0.05 | 0.621 | -0.04 | 0.633 |
| | DL | -0.006 | 0.925 | -0.03 | 0.792 | 0.006 | 0.951 |
| FAI-T (AU) | | | | | | | |
| | Sex | -0.02 | 0.814 | NA | NA | NA | NA |
| | Age (years) | 0.15 | 0.057 | 0.19 | 0.067 | 0.12 | 0.234 |
| | IOP (mmHg) | 0.17 | 0.004** | 0.17 | 0.05 | 0.22 | 0.005** |
| | cpRNFLT (μm) | 0.47 | <0.001*** | 0.54 | <0.001*** | 0.46 | <0.001*** |
| | Disc area | -0.16 | 0.009** | -0.21 | 0.021* | -0.12 | 0.125 |
| | MAP (mmHg) | -0.21 | 0.007** | -0.01 | 0.914 | -0.26 | 0.015* |
| | BAP (μmol/L) | 0.04 | 0.512 | 0.37 | <0.001*** | -0.19 | 0.040* |
| | Smoking history | 0.12 | 0.101 | 0.08 | 0.477 | 0.08 | 0.365 |
| | HT | -0.003 | 0.964 | 0.03 | 0.803 | -0.07 | 0.517 |
| | DM | 0.004 | 0.958 | 0.05 | 0.656 | 0.03 | 0.768 |
| | DL | -0.1 | 0.15 | -0.11 | 0.294 | -0.13 | 0.194 |

MBR-T: Mean blur rate in the tissue area of the optic nerve head; FAI-T: Flow acceleration index in the tissue area of the optic nerve head; AU: Arbitrary units; IOP: Intraocular pressure; cpRNFLT: Circumpapillary retinal nerve fiber layer thickness; MAP: Mean arterial pressure; BAP: Biological antioxidant potential; HT: Hypertension; DM: Diabetes; DL: Dyslipidemia; β: Standardized coefficient (linear mixed-effect model). Asterisks indicate statistical significance (*: $p < 0.05$, **: $p < 0.01$, ***: $p < 0.001$).

β = -0.08, p = 0.368). FAI-T was also associated with cpRNFLT in all the groups (overall, β = 0.47, p < 0.001; male, β = 0.54, p < 0.001; female, β = 0.46, p < 0.001). Disc area was also significantly associated with FAI-T in the overall group and the male group (overall, β = -0.16, p = 0.009; male, β = -0.21, p = 0.021). Although the association of disc area with FAI-T did not reach statistical significance in the female subjects (β = -0.12, p = 0.125), the direction of β was similar to the overall group and the male group. IOP was associated with FAI-T in the overall group and the female group (overall, β = 0.17, p = 0.004; male, β = 0.17, p = 0.050; female, β = 0.22, p = 0.005). Unlike MBR-T, MAP showed a significant association with FAI-T in the overall group and the female group (overall, β = -0.21, p = 0.007; male, β = -0.01, p = 0.914; female, β = -0.26, p = 0.015). Interestingly, a positive association between FAI-T and BAP was also found in the male group (overall, β = 0.04, p = 0.512; male, β = 0.37, p < 0.001; female, β = -0.19, p = 0.040).

## Discussion

In the current study, we investigated the relationship between OBF and oxidative stress markers and found that the serum BAP level was positively associated with MBR-T and FAI-T only

in male NTG patients. Although cpRNFLT is strongly associated with OBF in the ONH, our findings suggest that antioxidant capacity might be implicated in OBF impairment in NTG. Furthermore, there might be a sex difference in the role of oxidative stress in NTG.

## Oxidative stress and blood flow

Oxidative stress is defined as a disturbance in the balance between the production of reactive oxygen species (ROS) and antioxidants, and has been implicated in various diseases, including atherosclerotic cardiovascular disease (CVD) [16]. Although elevated IOP is the most significant risk factor for glaucoma, recent studies suggest that there is an association between oxidative stress and the pathophysiology of glaucoma. Oxidative stress can not only increase IOP via trabecular meshwork degeneration, but also directly damage the retinal ganglion cells through multiple cell death pathways [29,30] Furthermore, ROS can also induce vascular endothelial cell damage, which leads to blood flow impairment via decreased nitric oxide release [16,31]. Previously, we reported that increased urinary 8-OHdG levels were independently associated with decreased OBF in NTG [17], supporting the association between high oxidative stress and capillary damage in the ONH. However, we found only a poor correlation in the female subjects between OBF and the serum level of d-ROMs (rs = -0.18, p = 0.036), an indicator of systemic oxidative stress. Takayanagi et al. reported that serum d-ROMs were not correlated to the level of superoxide dismutase protein in the serum or aqueous humor or the level of serum sulfhydryl, and might not reflect the oxidative stress status of the eye [32]. Therefore, serum d-ROM levels might not directly reflect local ocular oxidative stress, which may explain the lack of an association between OBF parameters and d-ROMs in our study.

In the current study, we found that the serum BAP levels were positively associated with MBR-T in male patients with NTG. MBR represents a quantitative measurement of relative blood flow velocity. Reportedly, reduced MBR-T in the ONH precedes the loss of cpRFNLT and is independently associated with visual field loss in eyes with glaucoma, suggesting that MBR-T is a potential prognostic marker of glaucoma [13] [33]. Oxidative stress has been reported to enhance the inflammatory pathways responsible for the atherosclerotic process and to lower peripheral blood flow [34]. In addition, the acute oral intake of mitochondria-targeting antioxidants has been reported to contribute to improved endothelial function and peripheral circulation [35]. An animal study that used streptozotocin-induced diabetic rats showed that diabetic vascular dysfunction was accompanied by an accumulation of superoxides in arterioles, which normally provide blood flow to the sciatic nerve. That study also demonstrated that antioxidant treatment reduced vascular endothelial damage and improved peripheral nerve conduction [36]. Furthermore, decreased serum antioxidative capacity has been shown to increase oxidative stress in the aqueous humor of glaucoma patients [32]. Finally, oral antioxidant supplementation has been reported to increase ocular blood flow in the retinal and retrobulbar vascular beds in patients with open-angle glaucoma [37]. These previous studies support the findings of the current study, particularly that decreased antioxidant capacity was associated with decreased OBF, and suggest that the serum level of BAP is a useful biomarker of OBF impairment and underlying NTG pathogenesis. Thus, antioxidant supplements might be a promising treatment target for NTG patients with low serum BAP [38].

## Sex differences

Although various studies have reported findings that suggest lower antioxidant capacity is implicated in blood flow impairment, we observed an association between serum BAP and ONH OBF only in male NTG patients. Such sex differences have been noted in other diseases,

including CVD. Reportedly, women have higher mortality after acute cardiovascular events, though the incidence of CVD in women is usually lower than in men [39]. Yanagida et al. reported that women had better OBF than men among healthy volunteers, which is similar to our current findings. That study also suggested that the impact of age in decreased OBF was greater in women than men [40]. NTG has been reported to occur more frequently in women, and female sex has been reported to be a risk factor for the progression of NTG [41]. Vajaranant et al. reported that bilateral oophorectomy increased the risk of glaucoma [42], and Hulsman et al. found that reduced estrogen might be involved in blood flow impairment in women and that menopause was a risk factor for NTG [43]. Centofanti et al. reported that pre-menopausal women had significantly higher pulsatile ocular blood flow (POBF) and that estrogen replacement therapy in post-menopausal women improved POBF [44]. The secretion of estrogen varies among individuals, but it declines sharply during menopause at around age 50 [45,46]. Estrogen has been known to have anti-inflammatory and vasoprotective effects in addition to its role in stimulating follicles. Estrogen can modulate the expression of growth factors and oxidative stress, which is thought to partly explain its vasoprotective effects in injured arteries [47]. Moreover, the anti-inflammatory and vasoprotective effects of estrogen are lost in aging subjects. It has been also reported that blood total cholesterol, triglycerides, and low-density lipoprotein levels increase after menopause. Such unfavorable lipid profiles are involved in increased risk for CVD [48].

A recent systematic review and meta-analysis demonstrated that glaucoma patients had significantly higher total cholesterol, higher total cholesterol, and lower high-density lipoprotein levels than healthy controls [49]. Furthermore, hyperlipidemia can increase systematic oxidative stress in NTG patients [50]. Although we did not measure estrogen or cholesterol levels, these factors might affect OBF, resulting in sex differences in the association between antioxidant capacity and OBF in the ONH of NTG patients.

## Pulse waveform in the ONH and antioxidant capacity in glaucoma

The current study is the first to investigate an association between waveform parameters and antioxidant capacity in NTG patients. We used LSFG to measure OBF parameters and found that MBR-T and FAI-T were positively associated with BAP in male NTG patients. We also found that IOP and MAP were associated with FAI-T in the overall group of patients and the female patients. Iwase et al. demonstrated that FAI in the ONH significantly increased with increasing IOP in healthy subjects [51]. In addition, Yanagida et al. showed that FAI in the overall ONH was negatively associated with MAP [40]. These previous reports support our results showing the association of FAI-T with IOP and MAP. Although the male patients did not show a significant association between IOP or MAP and FAI-T, the direction of the effect size was similar to that of the female patients. The sex differences in the median of MBR-T and FAI-T might be involved in these results, but the exact reason could not be clarified by this study.

As already mentioned, it has been suggested that MBR-T is a good indicator of structural and functional changes in glaucoma [13,33]. FAI-T is an indicator of the instantaneous force that can increase OBF in a short time [25]. A case-control study of patients with European ancestry has shown that NTG patients have lower FAI-T (i.e., a flattened pulse waveform) than healthy subjects [52]. Curiously, the association between FAI-T and BAP in the current study showed different directions between the sexes (male β = 0.37, female β = -0.19). Although the exact reason remains unknown, several differences in background characteristics (e.g., younger age, lower BP, and higher d-ROM level) as well as sex hormone levels might be involved in the negative association of FAI-T with BAP in the female NTG patients. Moreover, the β was

low (β = -0.19) and the p-value was close to the threshold (p = 0.040), although it reached statistical significance. Therefore, the negative association of FAI-T with BAP in the female patients should be carefully interpreted. We previously reported that MBR decreases were milder in the ONH tissue of healthy subjects during IOP elevation, possibly because of increased FAI-T induced by the local autoregulatory system for OBF [53]. Therefore, antioxidant capacity might play an important role in OBF maintenance by preserving the autoregulatory system, as well as protecting retinal vessel structure.

## Limitations

Our study has several limitations. First, it was limited to Japanese patients with NTG and was conducted at a single institution. Therefore, a follow-up study is required to determine whether the results apply to patients of other ethnicities. Second, our study used a cross-sectional design, and we were thus unable to determine whether the reduction in OBF was the effect of decreased antioxidant capacity or was related to structural changes in the ONH due to glaucoma progression. Third, although we found an association between MBR-T and BAP only in the male patients, we did not have any molecular data to support this finding of a sex difference. Multiple studies have reported that female hormones could contribute to the differences in OBF between men and women [40,54]. The effectiveness of postmenopausal hormone replacement therapy for glaucoma has been controversial [55], which might be due to a lack of detailed data on the time of menopause and estrogen levels. Therefore, future studies that include the time of menopause and estrogen level, as well as antioxidant capacity, are required to evaluate the association of sex differences with the oxidative stress-related reduction of OBF in NTG.

In conclusion, we found that a lower serum antioxidant level, as indicated by BAP, was associated with reduced MBR-T in ONH only in male NTG patients. In addition, BAP showed a different association with FAI-T between the sexes. This finding suggests that oxidative stress might be an IOP-independent risk factor implicated in the pathogenesis of reduced OBF in NTG. Furthermore, sex differences should be taken into consideration for the management of NTG patients. Further study is required to verify the causal relationships of oxidative stress and OBF in NTG and to elucidate the mechanisms underlying sex differences in these relationships.

## Supporting information

**S1 Fig. Correlation of BAP with MBR-T and FAI-T.** MBR-T was significantly correlated with BAP only in the male patients (rs = 0.21, p = 0.036). FAI-T was significantly correlated with BAP in the overall group (rs = 0.16, p = 0.017) and the male patients (rs = 0.36, p < 0.036).
(TIF)

**S1 Data. All relevant data.** All relevant data are available in S1 Data.
(CSV)

## Acknowledgments

Involved in design and conduct of the study were (M.S. and T.N.); data collection (M.S. and N. T.); management and analysis (M.S., K.H., and M.Y.); interpretation of the data (M.S., N.H., and M.Y.); drafting of the manuscript (M.S. and M.Y.); and review and approval of the manuscript (T.N.).

We thank Mr. Tim Hilts for reviewing and editing the language of the manuscript.

## Author Contributions

**Data curation:** Nana Takahashi.

**Formal analysis:** Masataka Sato, Masayuki Yasuda, Kazuki Hashimoto.

**Supervision:** Toru Nakazawa.

**Validation:** Noriko Himori.

**Writing – original draft:** Masataka Sato, Masayuki Yasuda.

**Writing – review & editing:** Noriko Himori, Toru Nakazawa.

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
