## [Decision Letter · Decision Letter 0]

1 Nov 2022

PONE-D-22-21223Sex differences in the association between oxidative stress markers and optic nerve head blood flow in normal-tension glaucomaPLOS ONE

Dear Dr. Nakazawa,

Thank you for submitting your manuscript to PLOS ONE. After careful consideration, we feel that it has merit but does not fully meet PLOS ONE’s publication criteria as it currently stands. Therefore, we invite you to submit a revised version of the manuscript that addresses the points raised during the review process.

 Both reviewers found the MS of interest and suggested minor revisions. We look forward to the revised version 

We look forward to receiving your revised manuscript.

Kind regards,

Demetrios G. Vavvas

Academic Editor

PLOS ONE

Journal Requirements:

“This paper was supported in part by JSPS KAKENHI Grants-in-Aid for Scientific Research (B) (T.N. JP20H03838), Grant-in-Aid for Challenging Exploratory Research (T.N. JP21K19548), and Grant-in-Aid for Early-Career Scientists (M.Y. JP21K16866).

The funders had no role in the design or conduct of the study; collection, management, analysis, or interpretation of the data; preparation, review, or approval of the manuscript; or the decision to submit the manuscript for publication.”

Reviewers' comments:

Reviewer's Responses to Questions

**Comments to the Author**

1. Is the manuscript technically sound, and do the data support the conclusions?

Reviewer #1: Yes

Reviewer #2: Yes

2. Has the statistical analysis been performed appropriately and rigorously? 

Reviewer #1: Yes

Reviewer #2: Yes

3. Have the authors made all data underlying the findings in their manuscript fully available?

Reviewer #1: Yes

Reviewer #2: Yes

4. Is the manuscript presented in an intelligible fashion and written in standard English?

Reviewer #1: Yes

Reviewer #2: Yes

5. Review Comments to the Author

Reviewer #1: This work investigates the role of oxidative stress / antioxidant potential and optic nerve head perfusion parameters in normal tension glaucoma. The manuscript is well written, with interesting data for the scientific community but some revision is needed.

1) Lines 128-130: exclusion criteria “a history of high IOP (higher than 22 mmHg, as measured with Goldmann applanation tonometry) on the test day”. explain why patients with IOP of 22 mmHg were included, since this characterizes open angle glaucoma. Also, briefly explain why high myopia patients were excluded.

2) Line 154: “Measurements of ocular blood flow” This paragraph does not provide methods for measurement of ocular blood flow. Please review.

3) Lines 256-257 and 384-386: Authors state that ß value was very small / low. What criteria authors used to classify a ß value as very small/low? Other associations on the manuscript had similar ß value, but that information was not highlighted.

4) Lines 300-301: authors state no significant association between dROMs and OBF were found. However, authors also report a negative correlation between dROMs and FAI-T in females (rs=-0.18, p=0.036). Please review the statement.

5) Lines 305-306: If authors present previously published data showing that dROMs levels might not directly reflect local ocular oxidative stress, please explain why dROMs was used as oxidative stress marker in this study in the first place.

6) The main findings are related to BAP (antioxidant potential) and not dROMs (oxidative stress marker). BAP is not a direct oxidative stress marker, instead it reflects the systemic antioxidant potential by reducing ferric oxide to ferrous oxide. Therefore, the TITLE and ABSTRACT should be reworded to give less emphasis to “oxidative stress markers”.

Reviewer #2: This is a very interesting study investigating and reporting on the association of ocular blood flow parameters with oxidative stress markers in patients with normal tension glaucoma. More specifically, the study focuses on differences in these correlations that are observed between genders. The authors managed to present a sound and thoroughly organized study and relate their findings very well to previous studies. However, the following points need to be taken into consideration.

Abstract: The investigation of sex differences regarding the association of ocular blood flow with oxidative stress should also be mentioned in the purpose of the study.

The methods and results presented in the abstract should precisely reflect the investigations and outcomes of the present study. In particular, the authors should additionally refer to the outcomes regarding the correlation of MBR-T to d-ROMs, the other oxidative stress marker. A possible association of pulse-waveform parameters with oxidative stress markers and other clinical parameters was also investigated in the present study and should be mentioned both in methods and results of the abstract.

Line 44: The authors had better replace “MBR-T” with “mean blur rate in the tissue area of the optic nerve head (MBR-T)”, since the abbreviation has not been explained earlier.

Line 52: MBR-T was significantly correlated with disc area, too. The text should be changed accordingly.

Line 132: It is advisable that the authors explain why high myopia is included in the exclusion criteria.

Line 154: The title should be changed in order to correspond to the text that follows.

Line 167: The authors should further explain in more detail what the parameter “mean blur rate” exactly indicates.

Lines 171-172: References should be provided.

Table 2: Explanations for the abbreviations “BOS-T”, “BOT-T” should also be included in the caption of the table.

Lines 206, 212: Statical should be changed to statistical.

Line 228: FAI-T was significantly correlated with disc area too. The text should be changed accordingly.

Line 230: cpRNFLT was significantly correlated to FAI-T both in males and females. The text should also be changed accordingly.

Tables 3 and 4: The statistical method that was used for the presented analysis should be mentioned at the title of each table, i.e. spearman’s correlation coefficient and multivariate linear-mixed effect models, respectively. Furthermore, explanation for the abbreviation “AU” should be included in the caption of the tables.

Figure S1: A caption should be provided.

I would like to look at a revised version of the manuscript.

6. PLOS authors have the option to publish the peer review history of their article (what does this mean?). If published, this will include your full peer review and any attached files.

Reviewer #1: No

Reviewer #2: No

---

## [Author Response · Author response to Decision Letter 0]

14 Dec 2022

Response letter:

We wish to express our appreciation to the reviewers for their insightful comments on our paper. We feel the comments have helped us significantly improve our paper.

Journal Requirements:

Author’s response: Thank you for your comments. We have reviewed the author’s guidelines carefully and confirmed that our manuscript meets PLOS ONE's style requirements.

Author’s response: Thank you for your comments. In this study, the funders had no role. Thus, the statement below is correct:

We have added this statement to our cover letter, just in case.

Author’s response: All relevant data have been uploaded in the S1 Data file. We have added the following statement to the manuscript (line 691 in the revised version): “All relevant data are available in S1 Data.” We have also added this statement to the cover letter.

4. Please include captions for your Supporting Information files at the end of your manuscript, and update any in-text citations to match accordingly. 

Author’s response: We have added captions for “S1 Data” and “S1 Fig” at the end of our manuscript (lines 821-826).

“This paper was supported in part by JSPS KAKENHI Grants-in-Aid for Scientific Research (B) (T.N. JP20H03838), Grant-in-Aid for Challenging Exploratory Research (T.N. JP21K19548), and Grant-in-Aid for Early-Career Scientists (M.Y. JP21K16866).

The funders had no role in the design or conduct of the study; collection, management, analysis, or interpretation of the data; preparation, review, or approval of the manuscript; or the decision to submit the manuscript for publication.”

Author’s response:

We removed the funding-related text from the manuscript. The funding statement below is correct (we have also added this statement to our cover letter):

Author’s response: We have made some modifications to the reference list, as follows:

We deleted one of the citations (No. 31 in the original manuscript) because it was a duplicate of No. 13. In addition, we have added references (Nos. 23 and 24) in response to the reviewers’ comments. We also updated the citation numbers in the revised manuscript.

Comments to the Author

Reviewer #1

1) Lines 128-130: exclusion criteria “a history of high IOP (higher than 22 mmHg, as measured with Goldmann applanation tonometry) on the test day”. explain why patients with IOP of 22 mmHg were included, since this characterizes open angle glaucoma. Also, briefly explain why high myopia patients were excluded.

Author’s response: Thank you for pointing out the error in the description of the exclusion criteria. The subjects of the current study were normal-tension glaucoma patients with an IOP of less than 22 mmHg. Therefore, “a history of high IOP (higher than 22 mmHg)” in the exclusion criteria was incorrect. We have revised “higher than 22 mmHg” to “22 mmHg or higher” (p.8, line 164). 

The reason why we excluded high myopia patients was that the ocular circulation in myopic eyes has been reported to be different from normal eyes (PMID: 12084747, 12035987). In order to focus as closely as possible on ocular blood flow in normal-tension glaucoma, we excluded high myopia patients.

2) Line 154: “Measurements of ocular blood flow” This paragraph does not provide methods for measurement of ocular blood flow. Please review.

Author’s response: As the reviewer has pointed out, the title “Measurements of ocular blood flow” was inappropriate. We deleted this title and merged the paragraph with the next paragraph, “Optic nerve head blood flow assessment with laser speckle flowgraphy” (p 9-10, line 195-213).

3) Lines 256-257 and 384-386: Authors state that ß value was very small / low. What criteria authors used to classify a ß value as very small/low? Other associations on the manuscript had similar ß value, but that information was not highlighted.

Author’s response: We thank the reviewer for the comment. As pointed out, we mentioned no clear criteria for classifying the strength of ß values. However, an absolute correlation coefficient value of less than 0.2 has been judged to be a poor correlation in past reports, e.g., Acock et al, who defined |ß| < 0.2 as a weak effect size (Acock, A. C. 2014. A gentle introduction to Stata. College Station, Texas: Stata Press). Following this, we have described the ß value in line 433 as “very small” and the value in line 582 as “low” in the manuscript.

4) Lines 300-301: authors state no significant association between dROMs and OBF were found. However, authors also report a negative correlation between dROMs and FAI-T in females (rs=-0.18, p=0.036). Please review the statement.

Author’s response: As the reviewer has pointed out, the description “we did not find any significant correlation between OBF and the serum level of d-ROMs” is incorrect. In the Spearman correlation analysis, d-ROM level was correlated with FAI-T in the female subjects (rs=-0.18, p=0.036). However, although the p-value reached the significance threshold, the rs value was less than 0.20, which is considered to be a poor correlation (Rea, L. M., & Parker, R. A. 2014. Designing and conducting survey research: A comprehensive guide. San Francisco: Jossey-Basspoor).

Moreover, we found a only poor association between d-ROM level and FAI-T in a multivariate linear mixed-effect model for the female subjects (data not shown). Thus, we revised “we did not find any significant correlation between OBF and the serum level of d-ROMs” to “we found only a poor correlation in the female subjects between OBF and the serum level of d-ROMs (rs = -0.18, p = 0.036)” (p 22, lines 492-493).

5) Lines 305-306: If authors present previously published data showing that dROMs levels might not directly reflect local ocular oxidative stress, please explain why dROMs was used as oxidative stress marker in this study in the first place.

Author’s response:

The d-ROM level has advantages as a marker of oxidative stress for use in daily clinical practice, because it is easy to assess and can be measured simultaneously with BAP with a single analyzer. Therefore, we used d-ROMs as an oxidative stress marker. As the reviewer points out, however, d-ROM levels might not be a good indicator of local or ocular oxidative stress, as reported in previous papers (PMID: 33352680). We will investigate other oxidative stress markers for use in future analyses. We offer our thanks for the useful comment.

6) The main findings are related to BAP (antioxidant potential) and not dROMs (oxidative stress marker). BAP is not a direct oxidative stress marker, instead it reflects the systemic antioxidant potential by reducing ferric oxide to ferrous oxide. Therefore, the TITLE and ABSTRACT should be reworded to give less emphasis to “oxidative stress markers”.

Author’s response: We agree with the reviewer’s comment. In response, we revised “oxidative stress markers” to “systemic oxidative stress status” in the title.

Reviewer #2

Abstract: The investigation of sex differences regarding the association of ocular blood flow with oxidative stress should also be mentioned in the purpose of the study.

The methods and results presented in the abstract should precisely reflect the investigations and outcomes of the present study. In particular, the authors should additionally refer to the outcomes regarding the correlation of MBR-T to d-ROMs, the other oxidative stress marker. A possible association of pulse-waveform parameters with oxidative stress markers and other clinical parameters was also investigated in the present study and should be mentioned both in methods and results of the abstract.

Author’s response:

We thank the reviewer for the important suggestion. We have mentioned that a purpose of the study was to investigate sex differences in the association of ocular blood flow with oxidative stress.

As the reviewer has pointed out, the methods and results section in our abstract did not completely reflect the contents of the main manuscript. We revised the abstract to match the main manuscript as much as possible. However, we were unable to include all the results in the abstract because of the word-count limitation.

Line 44: The authors had better replace “MBR-T” with “mean blur rate in the tissue area of the optic nerve head (MBR-T)” since the abbreviation has not been explained earlier.

Author’s response: We thank the reviewer for pointing this out. We made the requested change (p 3, line 47).

Line 52: MBR-T was significantly correlated with disc area, too. The text should be changed accordingly.

Author’s response: We thank the reviewer for pointing this out. In response, we have added “and disc area (rs = -0.26, p < 0.001)” (p 3-4, lines 56-69).

Line 132: It is advisable that the authors explain why high myopia is included in the exclusion criteria.

Author’s response: As we also mentioned in our response to the comments of reviewer 1, the reason why we excluded high myopia patients was that the ocular circulation of myopic eyes has been reported to be different from normal eyes (PMID: 12084747, 12035987). In order to focus as closely as possible on ocular blood flow in normal-tension glaucoma, we excluded high myopia patients.

Line 154: The title should be changed in order to correspond to the text that follows.

Author’s response: Thank you for pointing this out. We removed the title “Measurements of ocular blood flow,” modified the relevant paragraph, and incorporated it into the next paragraph (p 9-10, lines 195-213).

Line 167: The authors should further explain in more detail what the parameter “mean blur rate” exactly indicates.

Author’s response: As the reviewer has suggested, we added an explanation of “mean blur rate” as follows (p 10, lines 215-216):

“...which is derived from the speckle pattern produced by the interference of a laser scattered by moving blood cells.”

We also have added a citation for this (No. 23).

Lines 171-172: References should be provided.

Author’s response: We have added a reference (No. 24).

Table 2: Explanations for the abbreviations “BOS-T”, “BOT-T” should also be included in the caption of the table.

Author’s response: We added expansions of the abbreviations “BOS-T” and “BOT-T” to the captions of Tables 2 and 3.

Lines 206, 212: Statical should be changed to statistical. 

Author’s response: We have changed “statical” to “statistical” in all tables.

Line 228: FAI-T was significantly correlated with disc area too. The text should be changed accordingly.

Author’s response: We left out a note on the correlation between FAI-T and disc area. In response to this comment, we added the following text to the manuscript: “and disc area (rs = -0.21, p = 0.010)” (p 15, line 304).

Line 230: cpRNFLT was significantly correlated to FAI-T both in males and females. The text should also be changed accordingly.

Author’s response: We have changed “cpRNFLT was significantly correlated to FAI-T both in males and females” to “CpRNFLT was significantly correlated to FAI-T in both sexes” (p 15, lines 307-308). 

Tables 3 and 4: The statistical method that was used for the presented analysis should be mentioned at the title of each table, i.e. spearman’s correlation coefficient and multivariate linear-mixed effect models, respectively. Furthermore, explanation for the abbreviation “AU” should be included in the caption of the tables.

Author’s response: We amended the title of Table 3 to “Spearman’s correlation coefficient between MBR-T, pulse waveform parameters, and each variable” and the title of Table 4 to “The effect of clinical parameters on MBR-T and FAI analyzed with a linear mixed effect model.”

We also added an expansion of the abbreviation “AU” to the captions of Tables 2, 3, and 4.

Figure S1: A caption should be provided.

Author’s response: We have provided a caption at the end of the manuscript, as follows:

“MBR-T was significantly correlated to BAP only in the male patients (rs = 0.21, p = 0.036). FAI-T was significantly correlated with BAP in the overall group (rs = 0.16, p = 0.017) and the male patients (rs = 0.36, p < 0.036).”

Additional change:

We deleted one of the citations (No. 31 in the original manuscript) because it was a duplicate of No. 13. In addition, we have added references (Nos. 23 and 24) in response to the reviewers’ comments. Accordingly, we updated the citation numbers in the revised manuscript.

In addition, we revised the expansions of the abbreviations at the end of each table.

---

## [Decision Letter · Decision Letter 1]

7 Feb 2023

Sex differences in the association between systemic oxidative stress status and optic nerve head blood flow in normal-tension glaucoma

PONE-D-22-21223R1

Dear Dr. Nakazawa,

We’re pleased to inform you that your manuscript has been judged scientifically suitable for publication and will be formally accepted for publication once it meets all outstanding technical requirements.

Kind regards,

Demetrios G. Vavvas

Academic Editor

PLOS ONE

Additional Editor Comments (optional):

Reviewers' comments:

Reviewer's Responses to Questions

**Comments to the Author**

1. If the authors have adequately addressed your comments raised in a previous round of review and you feel that this manuscript is now acceptable for publication, you may indicate that here to bypass the “Comments to the Author” section, enter your conflict of interest statement in the “Confidential to Editor” section, and submit your "Accept" recommendation.

Reviewer #1: All comments have been addressed

2. Is the manuscript technically sound, and do the data support the conclusions?

Reviewer #1: Yes

3. Has the statistical analysis been performed appropriately and rigorously? 

Reviewer #1: Yes

4. Have the authors made all data underlying the findings in their manuscript fully available?

Reviewer #1: Yes

5. Is the manuscript presented in an intelligible fashion and written in standard English?

Reviewer #1: Yes

6. Review Comments to the Author

Reviewer #1: Authors have adequately addressed all reviewers comments. The manuscript has improved overall. I have no further comments.

7. PLOS authors have the option to publish the peer review history of their article (what does this mean?). If published, this will include your full peer review and any attached files.

Reviewer #1: No

---

## [Editor Report · Acceptance letter]

15 Feb 2023

PONE-D-22-21223R1 

Sex differences in the association between systemic oxidative stress status and optic nerve head blood flow in normal-tension glaucoma 

Dear Dr. Nakazawa:

I'm pleased to inform you that your manuscript has been deemed suitable for publication in PLOS ONE. Congratulations! Your manuscript is now with our production department. 

Kind regards, 

on behalf of

Prof. Demetrios G. Vavvas 

Academic Editor

PLOS ONE